# Blue-wavelength light therapy for post-traumatic brain injury sleepiness, sleep disturbance, depression, and fatigue: A systematic review and network meta-analysis

Karan Srisurapanont[1], Yanisa Samakarn[1], Boonyasit Kamklong[1], Phichayakan Siratrairat[1], Arina Bumiputra[1], Montita Jaikwang[1], Manit Srisurapanont[2]*

1 Faculty of Medicine, Chiang Mai University, Chiang Mai, Thailand, 2 Department of Psychiatry, Faculty of Medicine, Chiang Mai University, Chiang Mai, Thailand

* manit.s@cmu.ac.th

## Abstract

### Objective

This review aimed to determine the efficacy of blue-wavelength light therapy (BWLT) for post-traumatic brain injury (TBI) sleepiness, sleep disturbance, depression, and fatigue.

### Methods

Pubmed, Scopus, Web of Science, Cochrane Library, Academic Search Complete, and CINAHL. Included trials were randomized controlled trials (RCTs) of BWLT in adults with a history of TBI. Outcomes of interest included sleepiness, sleep disturbance, depression, or fatigue. Two reviewers independently screened the searched items, selected the trials, extracted the data, and rating the quality of trials. We aggregated the data using a random-effect, frequentist network meta-analysis (NMA).

### Results

We searched the databases on July 4, 2020. This review included four RCTs of 117 patients with a history of TBI who were randomized to received BWLT, amber light therapy (ALT), or no light therapy (NLT). Moderate-quality evidence revealed that: i) BWLT was significantly superior to NLT in reducing depression (SMD = 0.81, 95% CI = 0.20 to 1.43) ii) BWLT reduced fatigue at a significantly greater extent than NLT (SMD = 1.09, 95% CI = 0.41 to 1.76) and ALT (SMD = 1.00, 95% CI = 0.14 to 1.86). Low-quality evidence suggested that BWLT reduced depression at a greater extent than ALT (SMD = 0.57, 95% CI = 0.04 to 1.10). Low-quality evidence found that the dropout rates of those receiving BWLT and ALT were not significantly different (RR = 3.72, 95% CI = 0.65 to 21.34).

### Conclusion

Moderate-quality evidence suggests that BWLT may be useful for post-TBI depression and fatigue.

**Data Availability Statement:** All data are in the Supporting Information Files: S1 Data. data.light. xlxs S1 Script. r.script.data.light.20210114.R

**Funding:** This work was supported by a grant from Chiang Mai University, Chiang Mai, Thailand (grant no. 16/2563 for M.S.). The funders had no role in study design, data collection and analysis, decision to publish, or preparation of the manuscript.

**Competing interests:** The authors have declared that no competing interests exist.

## Introduction

Traumatic brain injury (TBI) is common, increasingly prevalent, and one of the leading causes of disability worldwide. In 2016, there were 27·million new cases of TBI with an age-standardized incidence rate of 369 per 100 000 population [1]. As a medical condition related to road traffic injuries, TBI is more prevalent in low- and middle-income countries, especially in Africa and Southeast Asia [2].

Long-term behavioral sequelae of TBI play a crucial role in causing disability. Although the pathophysiology of post-TBI behavioral symptoms is complex and not fully understood, several studies have reported that (excessive daytime) sleepiness, sleep disturbance, depression, and fatigue are common in patients with a TBI. One to two years post-TBI, 43%-67% of the patients still have these behavioral symptoms [3–5]. Post-TBI sleepiness, sleep disturbance, depression, and fatigue seriously undermine patient rehabilitation, recovery, community reintegration, well-being, and quality of life [6–8].

Post-TBI depression, fatigue, and sleepiness are a symptom cluster similar to depressive disorders. In patients with major depressive disorder, fatigue is related to sleepiness and depression severity [9]. These three symptoms are similar to the core features of winter depression, including depression, low energy, and hypersomnia [10]. The evidence to date supports the use of bright light therapy as a treatment of choice for winter depression—a type of seasonal affective disorder [11]. It is, therefore, of interest to evaluate the efficacy of bright light therapy for post-TBI depression, fatigue, and sleepiness.

The exposure to blue- (or short-) wavelengths of light can affect subjective feelings of alertness and circadian physiology. Bright light therapy has physiological effects by resynchronizing the biological clock (circadian system), enhancing alertness, and acting on serotonin, and other monoaminergic pathways [12]. Morning exposure to bright light therapy can suppress nocturnal melatonin secretion from the pineal body resulting in the reduction of excessive daytime sleepiness and insomnia at night, as well as improving alertness [12–14]. Blue and red light are at the opposite ends of the visible spectrum. They characteristically have short wavelengths with high energy and long wavelengths with low energy, respectively. With a higher level of energy, blue-wavelength light would imply a natural preference to other wavelengths of light for the treatment of depression, fatigue, sleepiness, and sleep disturbance. Moreover, previous studies have shown that non-visual behavioral responses are highly sensitive to short-wavelength light [15, 16].

Many light therapy devices have been developed to treat medical conditions related to circadian rhythm disturbance. Of many devices, lightboxes are an electronic device commonly used for light therapy. However, they are not considered as medical devices and not regulated by an agency responsible for food and drug administration. The general requirements of a lightbox are clinical efficacy, ocular and dermatologic safety, and visual comfort [17].

More interventions remain needed for the treatment of post-TBI sleepiness, sleep disturbance, depression, and fatigue. Only stimulants and strategic use of caffeine/nap are possible treatment options for post-TBI sleepiness [18]. Although there is some evidence to support the benefits of antidepressants and/or cognitive-behavioral therapy for post-TBI depression, there is no well-established treatment for this particular depression [19]. A recent systematic review found little evidence to support the efficacy of aquatic physical activity, mindfulness-based stress reduction, and computerized working-memory training for fatigue [20]. However, this review did find a small-sample randomized controlled trial (RCT) reporting the benefits of BWLT for post-TBI fatigue [21].

Based on the rationales above and the need for more treatment options to manage these behavioral symptoms, we proposed to conduct a systematic review and meta-analysis of

randomized controlled trials (RCTs) to determine the efficacy of and adherence to BWLT for post-TBI sleepiness, sleep disturbance, depression, and fatigue.

## Materials and methods

The protocol of this systematic review was prospectively registered at Open Science Framework (https://osf.io/yf2qe/). The data (S1 Data) and script (S1 Script) are available as supporting information of this article. At first, we planned to conduct a pairwise meta-analysis. However, we found some difficulties in synthesizing the data comparing BWLT with two control interventions and a three-arm trial. Because a network meta-analysis (NMA) can validly compare multiple treatments across trials, we decided to conduct a NMA to aggregate these data. The report of this review followed the PRISMA Network Meta-Analysis Checklist [22]. For all review tasks independently conducted by two reviewers, any discrepancy was resolved using a consensus discussion with a third-party reviewer.

### Eligible criteria

The eligibility criteria for an included trial were as follows: i) a RCT; ii) adult participants (> 18 years old) with a history of TBI; iii) an experimental intervention of BWLT; iv) a control intervention of any kind of light therapy, e.g., amber light therapy (ALT), as well as no light therapy (NLT); and v) an outcome of sleepiness, sleep disturbance, depression, or fatigue.

### Information sources, searches, and trial selection

We performed database searches for eligible trials in Pubmed, Scopus, Web of Science, Cochrane Library, Academic Search Complete, and CINAHL Complete from their inceptions to July 4th, 2020. Key search terms included (light OR phototherapy) AND (((head OR brain) AND (trauma* OR injur*)) OR concussion) AND random*.

Neither language restriction nor publication date limitation was applied. Two reviewers (KS and YS) independently screened the titles/abstracts, evaluated the full-text publications to select the trial, and assessed the trial quality.

### Data collection process and data extraction

Two reviewers (KS and PS) independently extracted the trial data using a data record form. Characteristics of interest for each trial included: i) study ID (first author, year); ii) participant characteristics, including age, inclusion criteria, gender, diagnostic procedure(s), and details of the traumatic brain injury (e.g., Glasgow Coma Score); iii) details of light therapy and control intervention, including wavelength; iv) management guidelines and additional intervention (s); v) per-protocol analysis of the outcomes; and vi) the measures used for outcome assessment.

### Data items

The primary outcome of efficacy was sleepiness. The secondary outcomes included sleep disturbance, depression, and fatigue. For each outcome, we focused on the mean change scores and the SDs. The change scores were preferred because they could remove the between-person variability from the analysis, which results in greater efficient and more statistical power for detecting the differences [23]. Dropout rates were the measure of adherence to treatment. These data, as well as the number of participants for analyses, were extracted from the published articles.

For a missing standard deviation, we applied a sequence of strategies as follows: i) using the calculator of RevMan 5.4, ii) assuming that the standard deviations of all comparison groups were equal and using the mean and effect size for calculation, iii) contacting the corresponding author of that particular trial, iv) using the endpoint standard deviation of the particular group and outcome, and iv) measuring the graphs using the Engauge Digitizer version 6.0 software.

The extracted outcomes were used for computing the standardized mean differences (SMDs) and relative risks (RRs), as well as 95% confidence intervals (CIs).

## Geometry of the network

We compared the SMDs and RRs between interventions directly and indirectly. In the networks, an intervention was drawn by a node, and a line showed a comparison between the interventions. The line thickness indicates the number of trials and comparisons, respectively.

## Risk of bias within individual trials

Two reviewers (KS and YS) independently assessed the quality of included trials using the RoB2—a revised Cochrane risk-of-bias tool for randomized trials [24]. The bias tool evaluated five aspects of the trial as follows: i) randomization process, ii) adherence to the assigned intervention(s), iii) missing outcome data, iv) the bias of measurement, and v) the bias of the reported results. The levels of bias obtained from these aspects were used to determine the overall bias of each trial.

## Summary measures and methods of analysis

For each pairwise comparison, we used the SMDs following weight reduction as a measure of efficacy. The positive SMDs indicate the superiority of BWLT against ALT or NLT. We conducted random-effect pairwise meta-analyses and frequentist network meta-analyses (NMA) on an intention-to-treat analysis to determine the efficacy of BWLT for post-TBI sleepiness, sleep disturbance, depression, and fatigue. We prepared league tables to rank the studied interventions. The SMDs of 0.2, 0.5, and 0.8 were considered as small, medium, and large effect sizes, respectively [25].

## Assessment of inconsistency

For each outcome, global inconsistency was visualized with forest plots and quantified using $I^2$ statistics [26]. A $I^2$ value of 75% to 100% was rated as high inconsistency. For the dataset with high inconsistency, we further explored both within- and between-design inconsistency. In addition, we also calculated the inconsistency between direct and indirect estimates of each comparison.

## Additional analysis and risks of bias across trials

For the outcome in which all trials compared the same pair of interventions, we conducted a pairwise meta-analysis on an intention-to-treatment analysis using a random-effect model.

We used the risks of bias within individual trials to plot the distribution of risk-of-bias judgments within each bias domain [27]. The distribution of bias was weighted based on the sample sizes of trials. Moreover, we assessed the risks of bias across trials using funnel plots to visualize the publication bias of each outcome [23].

## Rating the quality of evidence

Direct and indirect estimates were downgraded using the GRADE Approach for rating the quality of evidence (effect estimates) derived from the NMA [28]. The quality of each estimate

was rated as high, moderate, low, and very low. Because all estimates were derived from RCTs, they were started at high quality and were downgraded by one level for each of the following concerns: a) risk of bias–high, b) inconsistency–substantial or uncomputable, c) indirectness– intransivity, d) imprecision– 95% CI of an effect estimate including null effect, and e) publication bias–high or uncomputable ($< 10$ trials). If only direct or indirect estimate was available for a given comparison, the NMA estimate was based on that estimate. For a particular comparison in which both direct and indirect estimates were available, the higher quality of the two quality ratings was used as the quality rating for the corresponding NMA estimate. The quality of each NMA estimate could be upgraded by one level for the presence of a large magnitude of effect, dose-dependent gradient, or plausible confounding [29].

## Statistical software

The NMAs were carried out using the *netmeta* version 1.2–1 package [30]. The risk of bias was visualized using the *robvis* version 0.3.0 package [27]. The meta-analysis was performed using the *meta* version 4.12–0 package [31]. All packages were used under the R Program version 3.6 via the Rstudio software version 1.2.5 [32, 33].

# Results

## Trial selection

The database searches retrieved 1,697 items (see Fig 1 and S1 Text). After the duplicate removal, 1,034 records remained for title and abstract screening. We assessed 19 full-text publications and, finally, included four RCTs in this systematic review [21, 34–36].

## Network structure and geometry

Fig 2 shows the network graphs comparing BWLT (4 trials, n = 53), ALT (3 trials, n = 42), and NLT (2 trials, n = 20) for sleepiness, sleep disturbance, depression, and fatigue. The comparisons included BWLT vs. ALT (3 trials), BWLT vs. NLT (2 trials), and ALT vs. NLT (2 trials). Among four included trials, a three-arm trial compared BWLT, ALT, and NLT, and three two-arm trials compared BWLT vs. ALT or BWLT vs. NLT. The networks including in the NMAs were as follows: i) sleepiness (6 pairwise comparisons of 3 treatments in 4 RCTs) (see Fig 2A); ii) sleep disturbance and depression (5 pairwise comparisons of 3 treatments in 3 RCTs) (see Fig 2B); and iii) fatigue (4 pairwise comparisons of 3 treatments in 2 RCTs) (see Fig 2C).

## Trial characteristics

This systematic review included 117 patients with a history of TBI participating in four RCTs (see Table 1). The trials were conducted in patients with mild TBI (2 RCTs), severe TBI (1 RCT), and mixed-severity of TBI (1 RCT). Brain injury had occurred between 6.75 months and 9 years prior to the study enrolment. Except for a three-arm trial conducted by Sinclair and colleagues, the other three trials were two-arm trials. The wavelengths of blue light therapy and ALT used ranged between $\lambda_{max} = 465$ nm and $\lambda_{max} = 480$ nm and ranged between $\lambda_{max} = 530$ nm and $\lambda_{max} = 578$ nm. While three trials provided light therapy using a LED lightbox [21, 34, 36], one trial gave light therapy using a face-mounted device resembling glasses [35]. All instances of light therapy were self-administered at home in the morning.

 Measures used for assessing the outcomes were as follows: i) Epworth Sleepiness Scale (ESS) for sleepiness [37], ii) Pittsburgh Sleep Quality Index (PSQI) for sleep disturbance [38], iii) Beck Depression Inventory (BDI) and Hamilton Rating Scale for Depression (HRSD-17) [39, 40], and iv) Fatigue Severity Scale (FSS) for fatigue [41].

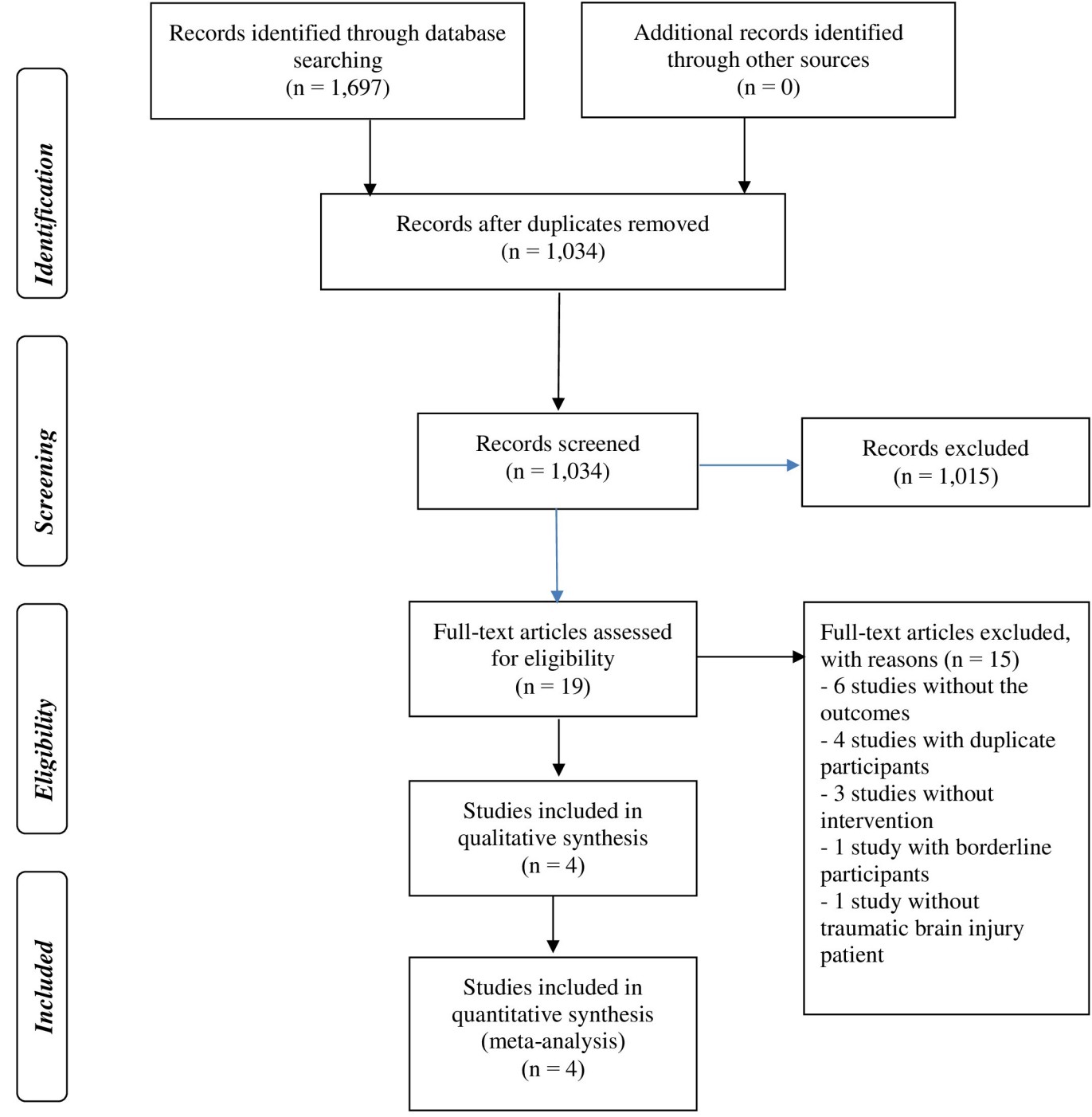

**Fig 1. PRISMA flow from literature search to study inclusion.**

## Risk of bias in individual trials

There were some concerns regarding the deviation from the intended interventions in all trials because BWLT was self-administered at home (see S1A Fig). No included trial was judged to be at high risk of bias.

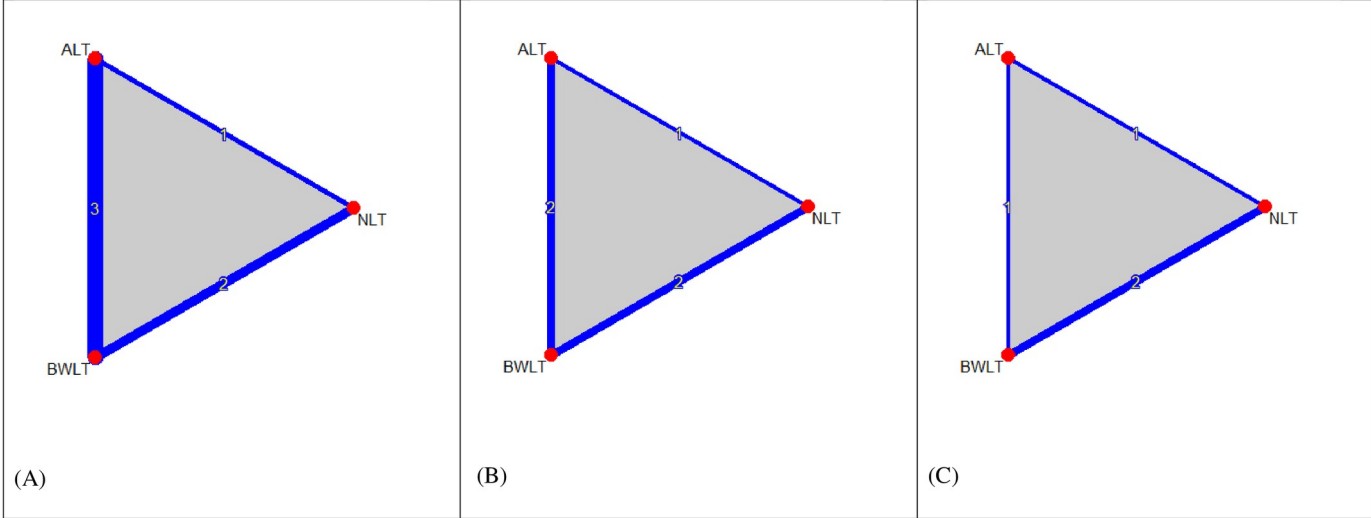

**Fig 2.** Network graphs of light therapy for behavioral symptoms of post-traumatic brain injury (A) sleepiness (6 pairwise comparisons of 3 treatments in 4 RCTs); (B) sleep disturbance and depression (5 pairwise comparisons of 3 treatments in 3 RCTs); (C) fatigue (4 pairwise comparisons of 3 treatments in 2 RCTs). Nodes in the graph layout correspond to the types of light therapy, and edges display the observed treatment comparisons. Edge thickness and the numbers on edges indicates the number of comparisons. Gray triangles display the presence of three-arm comparisons. ALT: Amber Light Therapy; NLT: No Light Therapy; BWLT: Blue-Wavelength Light Therapy.

## Results of individual trials

Of the four trials that assessed the reducting of sleepiness, three trials found the superiority of BWLT against ALT, one trial reported the superiority of BWLT against NLT, and one trial reported no difference between BWLT and NLT (see Table 1). All three trials that assessed the reducting of sleep disturbance reported no difference among BWLT, ALT, and NLT. Of the three trials reported the reduction of depression, two trials found the superiority of BWLT against ALT, but one trial found no difference among BWLT, ALT, and NLT. Both trials that assessed the reduction of fatigue reported the superiority of BWLT against ALT and NLT. Two trials reported the dropout rates of BWLT and ALT groups but did not test the statistical difference between groups.

## Synthesis of results

Fig 3A shows the forest plot of pooled SMDs comparing the efficacy of ALT and NLT against BWLT in reducing sleepiness. This forest plot revealed no significant difference between groups. However, a direct estimate of sleepiness reduction in the league table revealed a significant superiority of BWLT against ALT (SMD = -1.52, 95% CI = -2.98 to -0.07) (see Table 2A).

Fig 3B shows the forest plot of pooled SMDs comparing the efficacy of ALT and NLT against BWLT in reducing sleep disturbance. This forest plot revealed no significant difference between groups. Moreover, all direct and indirect estimates of sleep-disturbance reduction in the league table were not significantly different among these interventions (see Table 2B).

Fig 3C shows the forest plot of pooled SMDs comparing the efficacy of ALT and NLT against BWLT in reducing depression. BWLT was significantly superior to ALT (SMD = 0.57, 95% CI = 0.04 to 1.10) and NLT (SMD = 0.81, 95% CI = 0.20 to 1.43). In addition, the direct estimates of depression reduction in the league table also revealed the significant superiority of

**Table 1. Characteristics of four studies included in the systematic review.**

| Study (country, study duration) | Characteristics of the participants | Interventions (n) | Outcomes (results)[a,b,c] |
|---|---|---|---|
| **Sinclair 2014 (Australia, 4 weeks)** | • Patients with a history of mixed severity of TBI, who had significant fatigue or sleep problems | 1. BWLT ($\lambda_{max}$ = 465 nm, delivered via a lightbox) (n = 10) | Sleepiness: ESS (BWLT > ALT and NLT) |
| | | | Sleep disturbance: PSQI (BWLT $\cong$ ALT $\cong$ NLT) |
| | • % Male: 83.3 | 2. ALT ($\lambda_{max}$ = 574 nm) (n = 10) | Depression: BDI-II (BWLT $\cong$ ALT $\cong$ NLT) |
| | • Mean age: 42.0 years | | |
| | • Initial GCS: N/A | 3. NLT (n = 10) | Fatigue: FSS (BWLT > ALT and NLT) |
| | • Mean duration after TBI: 36.9 months | (For intervention 1 and 2: 45 minutes daily in the morning) | |
| **Quera Salva 2019 (France, 4 weeks)** | • Patients with a history of severe TBI, who had significant sleepiness, sleep disturbance, or fatigue | 1. BWLT ($\lambda_{max}$ = 468 nm, delivered via a face-mounted device resembling glasses) (n = 10) (30 minutes daily after waking) | Sleepiness: ESS (BWLT $\cong$ NLT) |
| | | | Sleep disturbance: PSQI (BWLT $\cong$ NLT) |
| | • % Male: 55 | | Depression: HRSD-17 (BWLT > NLT) |
| | • Mean age: 36.6 years | | |
| | • Initial GCS: 5.94 | | Fatigue: FSS (BWLT > NLT) |
| | • Mean duration after TBI: 9.03 years | 2. NLT (n = 10) | |
| **Killgore 2020 (USA, 6 weeks)** | • Patients with a history of mild TBI, who had sleep-related problems | 1. BWLT ($\lambda_{max}$ = 469 nm, delivered via a lightbox) (n = 18) | Sleepiness: ESS (BWLT > ALT) |
| | • % Male: 50 | 2. ALT ($\lambda_{max}$ = 578 nm) (n = 16) | Dropout rate (N/A) |
| | • Mean age: 23.7 years | (For both interventions: 30 minutes daily in the morning) | |
| | • Initial GCS: N/A | | |
| | • Mean duration after TBI: 6.75 months | | |
| **Raikes 2020 (USA, 6 weeks)** | • Patients with a history of mild TBI, who had disrupted sleep | 1. BWLT ($\lambda_{max}$ = 480 nm, delivered via a lightbox) (n = 17) | Sleepiness: ESS (BWLT > ALT) |
| | • % Male: 37.1 | | Sleep disturbance: PSQI (BWLT $\cong$ NLT) |
| | | | Depression: BDI-I: (BWLT > ALT) |
| | • Mean age: 25.9 years | 2. ALT ($\lambda_{max}$ = 530 nm) (n = 18) | Dropout rate (N/A) |
| | • Initial GCS: N/A | (For both interventions, 30 minutes daily in the morning) | |
| | • Mean duration after TBI: 9.20 months | | |

ALT: Amber Light Therapy; NLT: No Light Therapy; BWLT: Blue-Wavelength Light Therapy.

TBI: Traumatic brain injury; GCS: Glasgow Coma Scale; BDI-II: Beck Depression Inventory, 2[nd] edition; HRSD-17: 17-item Hamilton Rating Scale for Depression; FSS: Fatigue Severity Scale; ESS: Epworth Sleepiness Scale; PSQI: Pittsburgh Sleep Quality Index.

[a] Only the latest results of the study.

[b] The symbols of > and $\cong$ indicate significant superiority (p < 0.05) and not significant difference (p $\geq$ 0.05) as being reported by the authors, respectively.

[c] N/A indicates the report of data, but the statistical difference was not tested.

BWLT against ALT (SMD = -0.55, 95%CI = -1.09 to -0.01) and NLT (SMD = 0.77, 95% CIs = -1.42 to -0.12) (see Table 2C).

Fig 3D shows the forest plot of pooled SMDs comparing the efficacy of ALT and NLT against BWLT in reducing fatigue. BWLT was significantly superior to ALT (SMD = 1.00, 95% CI = 0.14 to 1.86) and NLT (SMD = 1.09, 95% CI = 0.41 to 1.76). Moreover, the direct estimates of fatigue reduction in the league table also revealed the significant superiority of BWLT against ALT (SMD = -1.00, 95%CI = -1.95 to -0.06) and NLT (SMD = -1.09, 95% CIs = f -1.76 to -0.41) (see Table 2D).

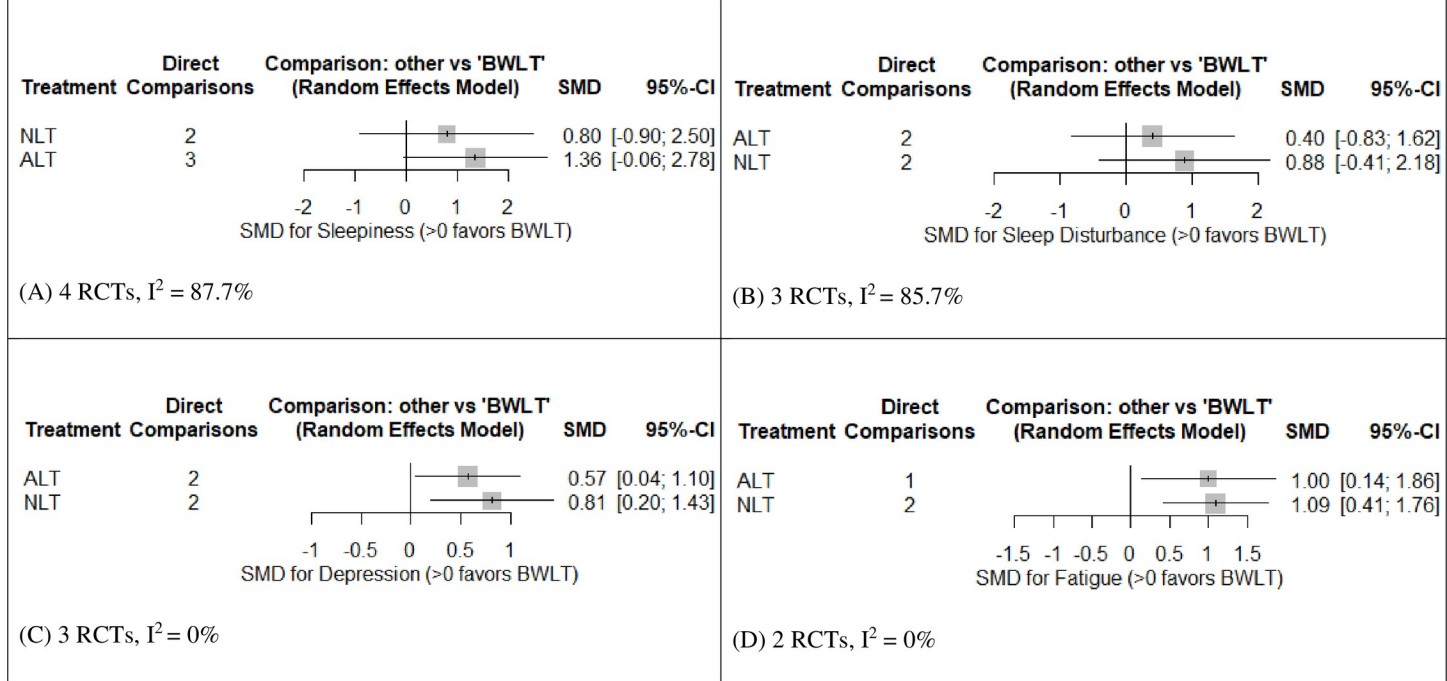

**Fig 3. Forest plots for the efficacy of ALT and NLT for behavioral symptoms of post-traumatic brain injury compared to BWLT.** (A) SMDs for the reduction of sleepiness; (B) SMDs for the reduction of sleep disturbance; (C) SMDs for the reduction of depression; and (D) SMDs the reduction of fatigue. SMDs > 0 indicate the superiority of BWLT against ALT or NLT. ALT: Amber Light Therapy; NLT: No Light Therapy; BWLT: Blue-Wavelength Light Therapy.

All direct, indirect, and NMA estimates for the comparisons of ALT and NLT found no significant difference in reducing any behavioral symptom.

## Inconsistency

The global inconsistency of sleepiness data was significantly high ($I^2$ = 87.4%, which was contributed by the inconsistency related to within- and between-study designs ($Q$ = 13.45, $df$ = 1, $p < 0.001$ and $Q$ = 10.41, $df$ = 2, $p = 0.006$, respectively). The global inconsistency of sleep-disturbance data was also significantly high ($I^2$ = 77.3%), which was contributed by the

**Table 2. League table presenting the treatment estimates and 95% confidence intervals of BWLT, ALT, and NLT for post-TBI sleepiness, sleep disturbance, depression and fatigue[a,b] - a random-effect model network meta-analysis estimates (lower triangle) and direct estimates (upper triangle)[c].**

| (A) SMDs for the reduction of sleepiness (95% CIs) | | | (B) SMDs for the reduction of sleep disturbance (95% CIs) | | |
|---|---|---|---|---|---|
| **BWLT** | -0.47 (-2.26 to 1.31) | **-1.52 (-2.98 to -0.07)** | **BWLT** | -0.15 (-1.42 to 1.11) | -0.98 (-2.32 to 0.37) |
| -0.80 (-2.50 to 0.90) | **NLT** | 0.57 (-1.95 to 3.08) | -0.40 (-1.62 to 0.83) | **ALT** | 0.20 (-1.64 to 2.04) |
| -1.36 (-2.78 to 0.06) | -0.56 (-2.49 to 1.38) | **ALT** | -0.88 (-2.18 to 0.41) | -0.49 (-1.99 to 1.02) | **NLT** |
| (C) SMDs for the reduction of depression (95% CIs) | | | (D) SMDs for the reduction of Fatigue (95% CIs) | | |
| **BWLT** | **-0.55 (-1.09 to -0.01)** | **-0.77 (-1.42 to -0.12)** | **BWLT** | **-1.00 (-1.95 to -0.06)** | **-1.09 (-1.76 to -0.41)** |
| **-0.57 (-1.10 to -0.04)** | **ALT** | -0.26 (-1.14 to 0.62) | **-1.00 (-1.86 to -0.14)** | **ALT** | -0.09 (-0.97 to 0.79) |
| **-0.81 (-1.43 to -0.20)** | -0.24 (-0.94 to 0.45) | **NLT** | **-1.09 (-1.76 to -0.41)** | -0.09 (-0.91 to 0.74) | **NLT** |

[a] SMDs less than 0 indicate the superiority of the light therapy defined in the column over the other light therapy defined in the row.

[b] Bold figures indicate significant differences between types of light therapy.

[c] Treatment are reported in order of ranking of efficacy. Comparison treatment should be read from left to right, and their standardized mean difference (SMD) in the cell in common between the column-defining light therapy and the row-defining light therapy.

| Study | BWLT Events | Total | ALT Events | Total | Risk Ratio | RR | 95%-CI | Weight (fixed) | Weight (random) |
|---|---|---|---|---|---|---|---|---|---|
| Killgore 2020 | 2 | 16 | 0 | 16 | | 5.00 | [0.26; 96.36] | 34.0% | 34.8% |
| Raikes 2020 | 3 | 17 | 1 | 18 | | 3.18 | [0.36; 27.65] | 66.0% | 65.2% |
| **Fixed effect model** | | **33** | | **34** | | **3.80** | **[0.66; 21.71]** | **100.0%** | **--** |
| **Random effects model** | | | | | | **3.72** | **[0.65; 21.34]** | **--** | **100.0%** |

Heterogeneity: $I^2 = 0\%$, $\tau^2 = 0$, $p = 0.81$

0.1 0.51 2 10
Dropout Rates (<1 favors BWLT)

**Fig 4. Forest plots for the dropout rates of BWLT compared with ALT in patients with a history of traumatic brain injury.** ALT: Amber Light Therapy; BWLT: Blue-Wavelength Light Therapy.

inconsistency related to between-study designs ($Q = 8.82$, $df = 2$, $p < 0.012$). The global inconsistency of depression and fatigue data was very low ($I^2$'s = 0%).

The analysis of Separate Indirect from Direct Evidence using back-calculation method found that all pairs of direct and indirect estimates had no significant inconsistency ($p$'s > 0.05).

### Additional analysis and risks of bias across trials

The dropout-rate data were obtained from two trials [34, 36]. Based on the pairwised meta-analysis, the pooled relative risk (95% CI) shows no significant difference between BWLT and ALT groups (RR = 3.72, 95% CI = 0.65 to 21.34, $I^2$ = 0%) (see Fig 4). The trials comparing BWLT and NLT did not report the dropout rates.

Weighted risks of bias across trials suggested that all trials had a risk of bias related to the deviations of intended interventions (see S1B Fig). The funnel plots of sleepiness, sleep disturbance, depression, fatigue, and dropout rate analyses revealed the variation of publication bias across outcomes (see S2A–S2E Fig). Because the numbers of comparisons were small, we did not apply any statistical method to test the bias.

### Rating the quality of evidence

The downgrading process of evidence quality was applied to direct and indirect estimates. These estimates were started at high quality because the NMA included RCTs only. The estimate quality was downgraded using the results mentioned above. The risk of bias was not concerned for these estimates because no included trial had a high risk of bias. The global inconsistency of sleepiness and sleep-disturbance data was high ($I^2$'s > 75%). The estimates related to these two outcomes were downgraded. Indirectness resulted in downgrading the estimate quality of all treatment outcomes because the participants and study designs were varied across trials (intransivity). The imprecision was based on each effect estimate (SMD or RR). The 95% CI of SMD including 0 or the 95% CI of RR including 1 would result in estimate downgrading. The publication bias of all outcomes could not be calculated. All estimates were downgraded by one level for this concern. The evidence quality of dropout-rate outcome was downgraded for two levels due to the imprecision and the undetermined publication bias. For the upgrading of NMA evidence, the NMA estimates larger than 0.8 were considered as those with a large magnitude of effect (large effect size) and were upgraded by one level.

Moderate-quality evidence revealed that: i) BWLT was significantly superior to NLT in reducing depression (SMD = 0.81, 95% CI = 0.20 to 1.43) ii) BWLT reduced fatigue at a

significantly greater extent than NLT (SMD = 1.09, 95% CI = 0.41 to 1.76) and ALT (SMD = 1.00, 95% CI = 0.14 to 1.86). Low-quality evidence suggested that BWLT reduced depression at a greater extent than ALT (SMD = 0.57, 95% CI = 0.04 to 1.10) (see Table 3).

Other NMA estimates showed no significant difference among BWLT, ALT, and NLT in reducing other behavioral symptoms. The low quality of evidence found that the dropout rates of those receiving BWLT and ALT were not significantly different (RR = 3.72, 95% CI = 0.65 to 21.34).

## Discussion

The findings of this systematic review and NMA should be viewed with caution because they were derived from a small sample of 117 patients who had a history of TBI and participated in four RCTs. Moderate-quality evidence suggested that BWLT was effective in reducing post-TBI depression and fatigue. The low and very low quality of evidence suggested that BWLT was not effective in reducing post-TBI sleepiness and sleep disturbance. The very low quality of evidence also suggested that ALT and NLT were not significantly different in reducing sleepiness, sleep disturbance, depression, and fatigue.

To our knowledge, this is the first quantitative synthesis of BWLT in reducing post-TBI sleepiness, sleep disturbance, depression, and fatigue. The efficacy of BWLT for post-TBI depression and fatigue found in this systematic review was in line with the results reported in individual trials included in this systematic review. Its efficacy for depression and fatigue has

**Table 3. Rating the quality of evidence.**

| | Direct estimate | | Indirect estimate | | NMA estimate | |
|---|---|---|---|---|---|---|
| | SMD (95% CI) | Quality of evidence | SMD (95% CI) | Quality of evidence | SMD (95% CI)[f] | Quality of evidence |
| Sleepiness | | | | | | |
| NLT: BWLT | 0.47 [-1.31; 2.26] | Very low [b-e] | 4.07 [-1.56; 9.69] | Very low [b-e] | 0.80 [-0.90; 2.50] | Very low |
| ALT: BWLT | 1.52 [0.07; 2.98] | Low [b,c,e] | -1.92 [-8.47; 4.63] | Very low [b-e] | 1.36 [-0.06; 2.78] | Moderate[g] |
| ALT: NLT | -0.57 [-3.08; 1.95] | Very low [b-e] | 2.19 [-0.84; 5.22] | Very low [b-e] | 0.56 [-1.38; 2.49] | Very low |
| Sleep disturbance | | | | | | |
| NLT: BWLT | 0.98 [-0.37; 2.32] | Very low [b-e] | -0.25 [-4.97; 4.47] | Very low [b-e] | 0.88 [-0.41; 2.18] | Low[g] |
| ALT: BWLT | 0.15 [-1.11; 1.42] | Very low [b-e] | 4.34 [-0.73; 9.41] | Very low [b-e] | 0.40 [-0.83; 1.62] | Very low |
| ALT: NLT | 0.20 [-1.64; 2.04] | Very low [b-e] | -1.89 [-4.52; 0.73] | Very low [b-e] | -0.49 [-1.99; 1.02] | Very low |
| Depression | | | | | | |
| NLT: BWLT | 0.77 [0.12; 1.42] | Low [c,e] | 1.25 [-0.78; 3.28] | Very low [c-e] | **0.81 [0.20; 1.43]** | Moderate[g] |
| ALT: BWLT | 0.55 [0.01; 1.09] | Low [c,e] | 1.01 [-1.32; 3.33] | Very low [c-e] | **0.57 [0.04; 1.10]** | Low |
| ALT: NLT | -0.26 [-1.14; 0.62] | Very low [c-e] | -0.22 [-1.36; 0.93] | Very low [c-e] | -0.24 [-0.94; 0.45] | Very low |
| Fatigue | | | | | | |
| NLT: BWLT | 1.09 [0.41; 1.76] | Low [c,e] | - | | **1.09 [0.41; 1.76]** | Moderate[g] |
| ALT: BWLT | 1.00 [0.06; 1.95] | Low [c,e] | 0.97 [-1.15; 3.10] | Very low [c-e] | **1.00 [0.14; 1.86]** | Moderate[g] |
| ALT: NLT | -0.09 [-0.97; 0.79] | Very low [c-e] | -0.06 [-2.53; 2.41] | Very low [c-e] | -0.09 [-0.91; 0.74] | Very low |

[a] risk of bias–no estimate downgrading (no trial with high risk of bias)

[b] inconsistency–estimate downgrading for sleepiness and sleep disturbance

[c] indirectness (intransivity)–estimate downgrading for all

[d] imprecision–estimate downgrading if 95% CI of SMD including 0

[e] publication bias–estimate downgrading for all

[f] Bold figures indicate significant differences between types of light therapy.

[g] large effect size–upgrading for NMA estimates > 0.8.

Estimates of effects and quality ratings for comparison of BWLT, ALT, and NLT for post-TBI sleepiness, sleep disturbance, depression and fatigue.

also been reported in other medical and psychiatric conditions. For example, a recent review suggested that bright light therapy was effective for seasonal and non-seasonal depression and depression in dementia [42]. Another systematic review also found that bright light therapy combined with psychosocial therapies was ranked the best in reducing cancer-related fatigue [43]. Taken together, bright light therapy, especially the BWLT, is effective in mitigating depressive and fatigue symptoms across the medical and psychiatric conditions, including the sequelae of TBI.

The inconclusive efficacy of BWLT for post-TBI sleepiness found in this systematic review is similar to previous findings of a RCT conducted in patients with Parkinson's disease [44]. However, the post-hoc analysis of that RCT still found the benefit of BWLT for sleepiness in a subgroup of patients with severe sleepiness (ESS score>11). Future trials in patients with moderate to severe sleepiness may be helpful to determine the therapeutic effects of BWLT for this post-TBI sleep problems.

The inconclusive efficacy of BWLT for sleep disturbance found in this systematic review is in line with the conflicting results of two previous reviews. While a review found the efficacy of BWLT for sleep problems in general, circadian rhythm sleep disorders, insomnia, and sleep problems related to Alzheimer's disease/dementia [45], the other study did not find such benefits in nursing home residents with sleep problems [46].

The high inconsistency of sleepiness and sleep-disturbance data ($I^2 > 75\%$) was in contrast with the homogenous data of depression and fatigue ($I^2 = 0\%$). Although it is difficult to determine the factors contributing to those two highly inconsistent outcomes, the data obtained from Killgore' trial might play a role. Among the four included trials, this trial reported only sleep outcomes. Moreover, this trial was conducted in the youngest participants (mean age of 23.7 years) who had the shortest duration after TBI (mean of 6.75 months).

The mechanisms of action of BWLT in reducing post-TBI depression and fatigue remain unknown. However, a recent study in healthy young adults found that morning exposure of BWLT was associated with a greater reduction of melatonin secretion compared with that of ALT [47]. Moreover, the BWLT also decreased the sleepiness and depressed mood to a greater extent than ALT. More evidence on the complex interactions among light, melatonin, mood, and sleep may help us understand more about the mechanisms of action of BWLT on mood and behavior not only in post-TBI but also in other mental and physical disorders.

Moderate-quality evidence found in this systematic review may remain insufficient to support the routine application of BWLT in patients with post-TBI depression and fatigue. However, as a relatively safe therapy with a large effect size of its efficacy, patients with post-TBI depression and fatigue may have a trial of BWLT with some cautions on its common adverse events of nausea, diarrhea, headache, and eye irritation [12]. The findings that ALT and NLT were not significantly different in reducing many behavioral symptoms might support the use of ALT as a placebo for the future trials of BWLT.

There were some limitations of this systematic review. First, this review included only four RCTs of 117 participants. For some outcomes, e.g., fatigue, dropout rates, the analyses included only 50 or 67 participants of two RCTs. A type II error, therefore, cannot be excluded in interpreting the inconclusive results regarding the benefits of BWLT on sleepiness and sleep disturbance. Second, the study duration of the included trials was relatively short (4–6 weeks) in comparison to the chronic nature of post-TBI sleepiness, sleep disturbance, depression, and fatigue. Future studies should have a longer study duration and include the assessment of symptoms several months after the end of treatment. Third, the participants included in this systematic review had a diverse history of TBI. While the severity varied from mild to severe, the mean duration after TBI ranged from 6.75 months to 9.03 years. This limitation might also contribute to the high inconsistency of some outcomes. Finally, the changed scores of many

outcomes in the trials included in this systematic review were not available. The requests for these data were unsuccessful. Although we tried to apply some methods to calculate the missing SDs, these might be less accurate than the actual ones.

## Conclusions

Based on moderate-quality evidence, BWLT may be useful for post-TBI depression and fatigue. The current evidence remains insufficient to support the routine application of BWLT for post-TBI depression and fatigue. Future studies in large sample sizes with longer study duration are warranted.

## Supporting information

**S1 Data.**
(XLSX)

**S1 Script.**
(R)

**S1 Table. PRISMA NMA checklist of items to include when reporting a systematic review involving a network meta-analysis.**
(PDF)

**S1 Text. Search strategies.**
(PDF)

**S1 Fig. Risks of bias assessed using the RoB (Risk of Bias) 2 tool for randomized trials.**
(PDF)

**S2 Fig. Funnel plots to visualize the publication bias of five outcomes.**
(PDF)

## Acknowledgments

We wish to thank Joan Peagam for her assistance with manuscript editing.

## Author Contributions

**Conceptualization:** Karan Srisurapanont, Yanisa Samakarn, Boonyasit Kamklong, Phichaya-kan Siratrairat, Arina Bumiputra, Montita Jaikwang, Manit Srisurapanont.

**Data curation:** Karan Srisurapanont, Yanisa Samakarn, Boonyasit Kamklong.

**Formal analysis:** Karan Srisurapanont, Manit Srisurapanont.

**Funding acquisition:** Manit Srisurapanont.

**Investigation:** Karan Srisurapanont, Yanisa Samakarn, Boonyasit Kamklong, Phichayakan Siratrairat, Arina Bumiputra, Montita Jaikwang, Manit Srisurapanont.

**Methodology:** Karan Srisurapanont, Yanisa Samakarn, Boonyasit Kamklong, Phichayakan Siratrairat, Arina Bumiputra, Montita Jaikwang, Manit Srisurapanont.

**Supervision:** Manit Srisurapanont.

**Writing – original draft:** Karan Srisurapanont, Yanisa Samakarn, Manit Srisurapanont.

**Writing – review & editing:** Karan Srisurapanont, Yanisa Samakarn, Boonyasit Kamklong, Phichayakan Siratrairat, Arina Bumiputra, Montita Jaikwang, Manit Srisurapanont.

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
