## [Decision Letter · Decision Letter 0]

13 Jan 2021

PONE-D-20-28037

Blue-wavelength light therapy for post-traumatic brain injury sleepiness, sleep disturbance, depression, and fatigue: a systematic review and network meta-analysis

PLOS ONE

Dear Dr. Srisurapanont,

Thank you for submitting your manuscript to PLOS ONE. After careful consideration, we feel that it has merit but does not fully meet PLOS ONE’s publication criteria as it currently stands. Therefore, we invite you to submit a revised version of the manuscript that addresses the points raised during the review process.

Based on the comments of the reviewers, we feel a minor revision would be appropriate.

We look forward to receiving your revised manuscript.

Kind regards,

Sahil Bajaj, Ph.D.

Academic Editor

PLOS ONE

Reviewers' comments:

Reviewer's Responses to Questions

**Comments to the Author**

1. Is the manuscript technically sound, and do the data support the conclusions?

Reviewer #1: Yes

Reviewer #2: Yes

2. Has the statistical analysis been performed appropriately and rigorously? 

Reviewer #1: Yes

Reviewer #2: Yes

3. Have the authors made all data underlying the findings in their manuscript fully available?

Reviewer #1: Yes

Reviewer #2: Yes

4. Is the manuscript presented in an intelligible fashion and written in standard English?

Reviewer #1: Yes

Reviewer #2: Yes

5. Review Comments to the Author

Reviewer #1: Thank you for the opportunity to review this manuscript. I do apologize for the delay in getting this review done. I know how frustrating it can be to sit and wait for reviewers to finish.

Developing and identify safe and efficacious TBI treatments is a growing public goal that is understandably burdened by generally small sample sizes in individual trials. To this end, systematic reviews/meta-analyses of this nature are essential to understanding the broader landscape of the literature.

Overall, I found this meta-analysis to be clearly written with clear methodology and interpretations that appropriately reflected the conducted analyses. However, I do have one concern that I did not find addressed anywhere in the manuscript. The four reviewed RCTs were conducted on very diverse populations. While all of the trials were on the TBI spectrum, Killgore and Raikes were exclusively on mTBI, while Sinclair was conducted on mixed severity and Quera Salva on severe. Additionally, treatment timings were very different across studies, with Killgore and Raikes being the most similar.

There is no indication that any of the analyses accounted for these very key differences. The longitudinal outcomes from severe TBI are very different from those for mild TBI/concussion. The recovery time curves are very different across these injury types as well. Particularly for a study like Quera Salva et al. where the average post-injury timing was 9 years compared to Raikes et al at 6 months, it's unclear how treatment timing and injury severity may play a role in these outcomes.

Please either clarify or revise this manuscript to address this concern. I recognize that there are only 4 studies to draw from and statistical methods are likely insufficient to tease apart these effects. The discussion should, at a minimum, present some interpretation/context for these issues.

Once again, thank you for the opportunity to read this manuscript. I look forward to the revised version.

Reviewer #2: I was delighted to review the manuscript. It addresses an interesting major topic (i.e. post-traumatic brain injury sleepiness, sleep disturbance, depression, and fatigue, and the efficacy of blue-wavelenght light therapy). The authors performed in a comprehensive methodologically way this systematic review and meta-analyses. The manuscript is nicely written and the results and discussion are coherent with the methodology. However, two minor issues need to be improved.

1. Introduction: First sentence ‘Post-TBI depression, fatigue, and sleepiness are…’ (page 3, line 66-67) and third sentence ‘These three symptoms are similar to…’ (page 3, line 68-69), could be rephrased to avoid repetition.

2. Information sources: ‘from their inceptions’ (page 6, line 133-134) it is better to include the date of the last search (e.g., from their inceptions to July 4th 2020).

Best wishes.

6. PLOS authors have the option to publish the peer review history of their article (what does this mean?). If published, this will include your full peer review and any attached files.

Reviewer #1: No

Reviewer #2: No

---

## [Author Response · Author response to Decision Letter 0]

14 Jan 2021

Thank you for the valued comments from reviewers and you. We have revised the manuscript accordingly as follows:

Reviewer 1:

1. The four reviewed RCTs were conducted on very diverse populations.

Our response: 

- We totally agree with this point. We have added three sentences in the limitations.

- Page 25, lines 466-469: Third, the participants included in this systematic review had a diverse history of TBI. While the severity varied from mild to severe, the mean duration after TBI ranged from 6.75 months to 9.03 years. This limitation might also contribute to the high inconsistency of some outcomes.

Reviewer 2:

1. Introduction: Avoid repetition - First sentence ‘Post-TBI depression, fatigue, and sleepiness are…

Our response:

- We revised the first sentence of this paragraph to cover the paragraph contents but not repeat the third sentence.

- Page 3, line 66-67: depressive disorders (replacing winter depression).

2. Information sources: change ‘from their inceptions’ to ‘from their inceptions to July 4th 2020’

- We revised the term accordingly

- Page 6, line 134: “from their inception to July 4th, 2020”.

---

## [Editor Report · Decision Letter 1]

15 Jan 2021

Blue-wavelength light therapy for post-traumatic brain injury sleepiness, sleep disturbance, depression, and fatigue: a systematic review and network meta-analysis

PONE-D-20-28037R1

Dear Dr. Srisurapanont,

We’re pleased to inform you that your manuscript has been judged scientifically suitable for publication and will be formally accepted for publication once it meets all outstanding technical requirements.

Kind regards,

Sahil Bajaj, Ph.D.

Academic Editor

PLOS ONE
---

## [Editor Report · Acceptance letter]

19 Jan 2021

PONE-D-20-28037R1 

Blue-wavelength light therapy for post-traumatic brain injury sleepiness, sleep disturbance, depression, and fatigue: a systematic review and network meta-analysis 

Dear Dr. Srisurapanont:

I'm pleased to inform you that your manuscript has been deemed suitable for publication in PLOS ONE. Congratulations! Your manuscript is now with our production department. 

Kind regards, 

on behalf of

Dr. Sahil Bajaj 

Academic Editor

PLOS ONE